# Confirmation of the Applicability of Skeletochronology and Estimating the Age Structure of *Kaloula borealis* (Microhylidae: Anura) at Lake Sihwa, South Korea

**DOI:** 10.3390/biology11060898

**Published:** 2022-06-10

**Authors:** Md Mizanur Rahman, Yu-Young Lee, Seung-Min Park, Choong-Ho Ham, Ha-Cheol Sung

**Affiliations:** 1Department of Biological Sciences, Chonnam National University, Gwangju 61186, Korea; mizanur32ju@gmail.com; 2Department of Biological Sciences·Biotechnology, Chonnam National University, Gwangju 61186, Korea; blue_lark@naver.com (Y.-Y.L.); parks5757@naver.com (S.-M.P.); hamch1007@hanmail.net (C.-H.H.)

**Keywords:** life history trait, growth curve, lines of arrested growth, innermost visible line, sexual size dimorphism

## Abstract

**Simple Summary:**

Age determination is very important for observing life history traits, evaluating vulnerable life stages, and setting proper management and conservation strategies. Age determination in animals has always been tricky. Among other approaches, skeletochronology, an age determination method that involves counting the lines of arrested growth (LAGs) produced during inactive periods, similar to plant year rings, is well-practiced in many animal groups. However, the applicability of this method remains questioned for many species, including amphibians, because of concerns around the scarcity of information on confirmed numbers of annual LAGs (e.g., once or twice in a year), chances of the disappearance of LAGs over time, and which lines exactly to count. Herein, we tested its applicability to *Kaloula borealis*, a class II endangered amphibian in South Korea, by rearing juveniles in the laboratory for more than one year and comparing the results with the wild population at Lake Sihwa. This study confirmed the formation of one LAG each year and no disappearance of LAGs over time in this species. Furthermore, we were also able to determine the age structure of this wild population accurately. Hence, our study validates using skeletochronology in this species and recommends it for others that show similar growth patterns.

**Abstract:**

Despite having some limitations, the use of skeletochronology—age determination by counting lines of arrested growth (LAGs)—in amphibians is increasing. The main limitation of using skeletochronology is identifying the innermost visible line (IVL) and counting the exact number of LAGs. Thus, we tested its applicability to *Kaloula borealis*, a class II endangered amphibian in South Korea. We reared juveniles in the lab to investigate the process of bone formation. This confirmed the development of one LAG each year. Hence, our study validates skeletochronology for the age determination of this species and recommends it for others that show similar growth patterns. Furthermore, the comparison of threshold diameters with the IVL of wild individuals confirmed no LAG1 resorption. The average age of males and females in this population was 2.75 ± 1.05 and 3.64 ± 3 years, respectively. We estimated sexual maturity at 2 years with rapid growth up to that stage in both sexes. We found a female-dominated sexual size dimorphism. This study offers accurate information on the life history traits and age structure of *K. borealis* that may help to evaluate population dynamics in other areas, identify vulnerable life stages and sites, assess the causes of population decline, and set conservation priorities.

## 1. Introduction

Knowledge of population dynamics and life history traits is key to understanding population decline and setting subsequent effective conservation strategies for amphibians [1,2,3]. Age structure may give an idea about a specific population’s dynamics and life history traits [4]. Skeletochronology, a method of counting annual growth lines, is a widely used and accepted method for estimating population age structure in aquatic and terrestrial animals [5,6,7,8]. Given the growth patterns of amphibians during hibernation, with a shrunken structure in the internal organs and a reduced growth rate [9], skeletochronology may also be highly applicable in determining amphibian population age structure [10].

However, the main limitation of using skeletochronology is identifying the innermost visible line (hereafter, IVL) and counting the exact number of lines of arrested growth (hereafter, LAGs) [10]. Generally, the cross-section of the bones contains an inner endosteal and an outer periosteal region differentiated by a sharp, almost circular line: Kastschenco’s line (hereafter, KL). With the aging of amphibians, the endosteal region enlarges outwards and resorption occurs for some parts of the periosteal region [10]. Thus, there is a possibility of LAG1 and/or LAG2 resorption and the underestimation of age [11,12,13,14,15]. Before endosteal resorption, the metamorphosis line (hereafter, ML), looks like LAG1, which may lead to overestimation when age counting [16]. To avoid these confusions and improve the method’s accuracy, Hemelaar (1985) proposed an effective method called the Back Calculation Method (hereafter, BCM) [13]. In this method, the IVL (the innermost periosteal line, immediately after the KL) is identified by comparing the LAG1 to that of an individual with a known age (after the first metamorphosis). Although the use of skeletochronology is increasing in determining the age structure of amphibian populations [8,17], many are not following BCM to make the study error-free.

Similar to other parts of the world, many Korean salamanders and anurans have also been investigated with skeletochronology [18,19,20,21,22,23,24,25]. Unfortunately, most of these studies with skeletochronology on Korean amphibians have not been able to identify the IVL. Among others, the Jeju population of *Kaloula borealis*, a class II endangered amphibian in South Korea (VU in South Korea [26] and LC from a global perspective [27]), has been included in these studies. Accordingly, the age structure of this population has remained a matter of debate. Considering the significance and scientific concerns about the population decline of this species in Korea, it is very important to understand the accurate age structure and growth patterns of this amphibian to make current and future restoration programs successful [28].

Thus, we reared and examined juveniles of *K. borealis* in the laboratory for more than one year (including the stages of metamorphosis, hibernation, and the post-hibernation active period) and studied a wild population at Sihwa Lake to evaluate the applicability of skeletochronology in determining the accurate age structure, life history traits, and growth patterns of this species. We mainly tried to address the following questions for this species in our study: (i) do LAGs represent annual growth? (ii) Can following the BCM identify IVL accurately? (iii) Is there any endosteal resorption in LAGs?

## 2. Material and Methods

### 2.1. Study Site

This study was conducted in Sinoe-ri, Namyang-eup, Hwaseong-si, Gyeonggi-do (N: 37°17′11.29″, E: 126°48′35.46″). The area is located on the eastern side of the Songsan Green City development project area, a segment of reclaimed land formed after the construction of the Sihwa Embankment. A river leading to Lake Sihwa flows to the northeast, and a large residential land development site with farmland and forest is located to the southwest (Figure 1). The habitat where the individuals were mainly found consisted of a shallow puddle temporarily formed during the rainy season, and in the herbaceous areas with sagebrush, fern, reed, etc.

### 2.2. Sample Collection

The samples of this study were collected during the 2013–2014 rainy seasons, i.e., early June to the end of August, which is the breeding period of *K. borealis*. Considering this species’ subterranean and nocturnal behavior, we conducted surveys mainly at night before and after high precipitation and high humidity when they become active and come out of hiding. Moreover, we used call recordings as bait to capture the individuals. For the identification of juveniles, we followed the Gosner developmental stage table [29]. We captured a total of ten juveniles in Gosner stage 47 (a state in which the front and hind legs are fully developed and the tail is completely absorbed).

The adult individuals (*n* = 136; 122 males, 14 females) were moved to the laboratory for the identification of sex and further observations. Sex identification was mainly based on external secondary sexual characteristics (the vocal sac under the chin and the presence of eggs on the stomach). In addition, we measured the snout to vent length (SVL) to the nearest 0.1 mm with digital calipers (Mitutoyo, CD-15CPX) and body weight (BW) to the nearest 0.1 g with a portable digital scale (Dongguan Hangchen, RE-500) before cutting the toe tips. After taking the measurements, the phalanges of each individual were cut for skeletochronology. Three phalanges (1st to 3rd) of the third toe of the forelimbs were taken and stored in 10% formalin (fixing solution). To prevent inflammation, the amputated toe part was disinfected with salve (iodin solution). After collecting the phalanges, the individuals were allowed to rest in a water tank (capture box) for about 30 min and then released to the capture point.

### 2.3. Rearing of Juveniles and the Determination of Reference ML and LAG1 Diameter

We reared the collected juveniles (immediately after metamorphosis in Gosner stage 47) in the Eco-Behav Lab, Department of Biological Sciences, Chonnam National University, South Korea, for more than a year to observe the bone formation process by growth stage and to obtain a standard diameter of ML and LAG1. We prepared the rearing cage (60 cm × 50 cm × 50 cm) with a 25–30 cm deep substrate (soil and leaf mold mixtures) and hollow logs and cork bark for hiding. Shallow water dishes and bowls were provided for their hydration. We fed them once a day with live mealworms and buffalo worms. High humidity (above 65%) and temperature (at 22–25 °C) were maintained with a humidifier and thermostat. We exposed them to a low temperature of 10 °C or less to gain artificially induced hibernation. In the reared individuals, hibernation periods varied from three to five months. We collected the phalanges and treated them with skeletochronology to see the bone formation process ‘immediately after the metamorphosis’ (20 days after metamorphosis), ‘before the first hibernation’ (at 18 weeks), ‘immediately after the first hibernation’ (36–41 weeks, depending on the individual hibernation period), and ‘before the next hibernation’ (at 68 weeks). However, we used the diameter of the ML and LAG1 for the juvenile bone cross-sections ‘after metamorphosis’ and ‘immediately after hibernation’ to obtain the threshold values for BCM.

For the identification of endosteal resorption and the IVL of the specimens, the mean ML and LAG1 values for juveniles (*n* = 10) were calculated following the BCM of Hemelaar (1985) and compared with the adult IVL mean diameter [13]. We calculated the mean diameter value of the reared juvenile MLs from the bone sections before and after hibernation. At the same time, the LGA1 of the same individuals was also calculated after hibernation. Four diaphyseal sections were selected from the middle phalanx collected from each individual, and the shortest axis and the longest axis of the IVL of each section were measured. After multiplying the long and short axis measurements together, we calculated the square root of the multiplied value. These obtained values were used to calculate the mean diameter of the IVL seen in each diaphyseal section. To determine a threshold and examine the endosteal resorption of ML and LAG1, we followed the ‘>2SD’ method [13,15]. We fixed a threshold diameter for ML and LAG1 from known-age individuals by adding 2SD to the mean values gained from the skeletochronology of reared juveniles (Appendix A).

### 2.4. Skeletochronology

The general histological staining method was followed to prepare phalanges for the skeletochronology process (Appendix A). The sections of the phalanges after histological staining were observed under an optical microscope (LEICA, DM500) at 400× magnification. We selected the sections at the mid-length of the diaphysis, where the marrow cavity is the narrowest, the periosteal bone is the widest, and the circle of the original bone age lines is best preserved [10]. After selecting ten diaphyseal sections from this part, pictures were taken with microscopy equipment (LEICA, ICC50HD) and edited with a microscopy program (LEICA, LAS Basic ver. 4.5) to better visualize the LAGs. We determined the age of an individual after the identification of the IVL by comparing it with the reference values of the reared juveniles. If the value exceeded the threshold of the ML, we considered it as a LAG1 and counted all lines as the age of that individual. If it did not exceed the threshold, we considered the IVL as an ML and deducted it when counting that individual’s age. We also looked for the endosteal resorption of LAG1 by comparing the IVL diameter with the threshold value of the reared individuals’ LAG1, but did not find any.

### 2.5. Estimation of the Growth Curve

To evaluate the relationship between age and growth rate, we estimated the growth curve for the Sihwa Lake *K. borealis* population. We used the individuals’ ages obtained by the skeletochronology and SVL data to generate the growth curve following von Bertalanffy’s equation [30]:*S_t_* = *S_m_* − (*S_m_* − *S*_0_)*e*
^−*K*(*t*−*t*_0_)^


Here, *S_t_* (average body length at age *t*) is the average individual size at *t* years of age, *S_m_* (asymptotic body length) is the maximum growth size, *S*_0_ (body length at metamorphosis) is the average size of an individual immediately after metamorphosis, *t* (age) is the age of the identified individual, *t*_0_ (age at metamorphosis) is the age of the individual until metamorphosis, and *K* (growth coefficient) is the growth coefficient.

### 2.6. Estimation of Sexual Dimorphism

The sexual dimorphism of the *K. borealis* population at Sihwa Lake was estimated following Lovich’s and Gibbons’ (1992) Sexual Dimorphism Index (SDI) [31]:*SDI* = (*size of larger sex/size of smaller sex*) ± 1


Here, (+1) was implemented if males were larger and (–1) if females were larger. The equation was arbitrarily defined as positive when females were larger than males, and negative for the contrary.

### 2.7. Statistical Analysis

To study the age characteristics of the breeding *K. borealis* population of Lake Sihwa, the collected data were subjected to a normality test (Kolmogorov–Smirnov normality test) before analysis. The relationship between age and body characteristics (SVL and BW) was analyzed using Spearman’s correlation analysis. Differences in SVL, BW, and age between years were analyzed using a nonparametric independent 2-sample test (Mann–Whitney). The diameter distribution of IVL in the breeding population was indicated through frequency analysis. All descriptive statistical data are presented as mean ± standard deviation (Mean ± SD). Data analysis was performed using SPSS (Statistical Package for the Social Sciences ver. 21), and the growth curve graph was indicated using the Sigmaplot program (SigmaPlot ver. 10.0).

## 3. Results

### 3.1. Bone Formation and the Appearance of LAG

Juveniles (*n* = 10) caught in the Sihwa Lake area were reared for more than one year, and the bone formation process was observed in the cross-section of the phalanges. The cross-section of the phalanges collected at the ‘immediately after the metamorphosis’ stage was characterized by an ML stained with hematoxylin in dark purple color with a big marrow cavity, a small periosteal, and a very narrow endosteal region (Figure 2A), whereas the periosteal bone grew widely out of the ML with a small endosteal part during the ‘before the first hibernation’ stage (Figure 2B). In this stage, the marrow cavity was partially filled by the endosteal bone and was narrower than in the ‘immediately after the metamorphosis’ stage.

In the cross-section of the phalanges at the ‘immediately after the first hibernation’ stage, in addition to ML, a conspicuous LAG1 was observed in the periosteal bone region (Figure 2C). In this stage, the periosteal and endosteal regions were enlarged, and the marrow cavity continued to be filled with endosteal bone, whereas the cross-sections of the phalanges of all individuals at the ‘before the next hibernation’ stage were characterized by a bigger periosteal region with LAG1 at the periphery and the ML towards the center. The endosteal region was larger than in previous stages and filled much of the marrow cavity. The marrow cavity was very small (Figure 2D). Thus, it is confirmed that LAGs are produced during hibernation periods once a year and represent year rings in this species.

### 3.2. Threshold Diameter for ML and LAG1

The mean diameter values of ML and LAG1 in the reared individuals (*n* = 10) were measured to derive the threshold diameter for them and to accurately determine the age in the breeding population of *K. borealis* in the Shihwa Lake area following the BCM. The mean diameter of ML was 109.75 ± 8.64 μm, and the mean diameter of LAG1 was 162.90 ± 8.63 μm. We calculated the threshold diameters of 127 μm and 180 μm for ML and LAG1, respectively (Table 1; Appendix A).

### 3.3. The Age Structure of the K. borealis Population in Lake Sihwa

We determined the age of the individuals in the *K. borealis* breeding population at Lake Sihwa through skeletochronology, counting the LAGs on the cross-sections of the phalanges. We observed the continuous concentric LAGs stained with hematoxylin. The IVL identity was verified by comparing it with the threshold diameter of the ML and LAG1 estimated in the laboratory-reared individuals. We found that the IVL had a mean diameter of 147 µm ± 12.20 (Figure 3F). After comparing with the standard diameter (ML and LAG1), the results revealed that the IVLs in the *K. borealis* population at Lake Sihwa were represented by LAG1 in most cases (94.85%; *n* = 129). Only a few individuals (5.15%; *n* = 7) had a ML as an IVL (Figure 3F). Furthermore, we did not find any endosteal resorption of LAG1 after comparing it with the standard LAG1 threshold obtained in the reared juveniles. We used the BCM to successfully identify the IVL in this species.

After determining the identity of IVL, we estimated the age of each individual by counting the LAGs (Figure 3A,B). We found the average age structure to be significantly higher in the female community than in the males (Table 2). Although the oldest breeding individual (8 years old) was a male, most of them were of only 2–4 years (96%; Figure 3C). The average age of the male community was 2.75 ± 1.05 years (*n* = 122; Table 2; Figure 3E), whereas the oldest adult female was 6 years and the youngest one was only 2 years. All of the age groups, except for the 6-year-old group, represented an almost equal proportion in the female community (Figure 3D). The average age of the female community was 3.64 ± 3 (*n* = 14; Table 2; Figure 3E). We estimated sexual maturity at two years in both males and females.

### 3.4. The Growth Curves

We found higher growth size and growth coefficient (K) values in females than in males in the *K. borealis* population (Table 3). The growth curves indicated a rapid growth up to 2 years of age in both sexes, and a slower growth after an age of 3 years (Figure 4).

### 3.5. Relationship of Age with Body Size and Weight

Spearman’s correlation analyses showed a positive correlation of age with the size and weight of the individuals in the *K. borealis* breeding population in Lake Sihwa (Figure 5). We calculated the correlation for the total period and each year separately to see whether there were any differences. We found a statistically significant positive correlation among the parameters of age, body size, and body weight across the whole period of our study. The values for the Spearman’s correlation coefficient of age with body size and body weight in the first year of our study were *r_s_* = 0.292 (*n* = 60, *p* < 0.023) and *r_s_* = 0.278 (*n* = 60, *p* < 0.031), respectively (Figure 5A,D), whereas in the second year, the values were *r_s_* = 0.596 (*n* = 76, *p* < 0.001) and *r_s_* = 0.616 (*n* = 76, *p* < 0.001), respectively (Figure 5B,E). Analyses for the total period gave the values *r_s_* = 0.502 (*n* = 136, *p* < 0.001) and *r_s_* = 0.520 (*n* = 136, *p* < 0.001) for the Spearman’s correlation coefficient of age with body size and body weight, respectively (Figure 5C,F).

### 3.6. Sexual Size Dimorphism

The SDI for *K. borealis* populations in Sihwa Lake was 0.087. This indicates that the females were larger than the males in this population. To ensure this trait in different stages of life history, we measured SDI for all age groups of the adult *K. borealis* individuals that we collected from the study site. The analyses revealed larger females in all age groups. The SDIs for 2 years, 3 years, 4 years, 5 years, and 6 years were 0.034, 0.055, 0.078, 0.189, and 0.085, respectively. We could not calculate SDIs for 7- and 8-year-old frogs as there were no female individuals of these age groups.

## 4. Discussion

We investigated the life history traits and sexual size dimorphism in different age groups of the breeding population of *K. borealis* at Sihwa Lake, South Korea. We used skeletochronology, a widely practiced age-determining method [32,33,34,35], to determine the population age structure in our study. We confirmed the formation of LAGs after every hibernation by rearing juveniles in the laboratory, and validated the applicability of skeletochronology in determining the age of *K. borealis* and other amphibians that undergo annual hibernation.

The laboratory rearing of the juveniles allowed us to observe the bone formation and growth patterns of *K. borealis*. Similar to Rozenblut and Ogielska (2005), the cross-sections of the bones in our study revealed readily differentiable regions of endosteal and periosteal with structures and staining patterns [10]. In accordance with previous studies [10], we found only MLs in the juvenile frog phalange cross-sections that were taken before the first hibernation, and both ML and LAG1 after hibernation. Furthermore, we did not observe the formation of any other lines resembling LAGs until the second hibernation of the next year. This indicates that hibernation causes a pause in growth and creates an annual growth ring (LAG1), which validates the applicability of skeletochronology in determining the age of this species, similar to many other species [11,12,36]. Considering the climatic conditions and hibernating nature of other Korean amphibians, we recommend skeletochronology as an effective method for determining age structures. We also recommend and urge the verification of the applicability of this method for other species that show similar growth patterns.

However, unlike some other studies [22,37], the appearance of MLs and LAG1s were similar. It was difficult to differentiate them except by the location: MLs towards the KL (endosteal bone) and LAG1s towards the periphery (periosteal bone). Many of the previous studies also reported similar results to ours [16]. Given the similarity in appearance of the ML and LAG1, it is very important to differentiate them and identify the IVL to avoid overestimating age and determine resorption possibilities [13,15]. In accordance with previous literature [13,14], we followed the BCM to identify and differentiate the ML and the LAG1, and found BCM as a useful method for identifying the IVL accurately in our study species. We found 94.85% of the IVLs were the LAG1, and only 5.15% were MLs in the *K. borealis* population at Sihwa Lake. In addition, we did not find any resorption of LAG1 in the studied population. In contrast to our findings, a few researchers suggest the continual resorption of LAGs (up to LAG1 and/or LAG2) [13,14,15], whereas Rozenblut and Ogielska (2005) and others support our results [10]. Thus, we oppose the possibility of age underestimation in skeletochronology for this species and suggest considering all visible lines after KL as a reference for LAG, except the lines which fail to pass the threshold diameter in the BCM (which should be considered as MLs). We strongly recommend the proper identification of IVL before determining the age of an individual and the age structure of a population.

The age structure of the *K. borealis* population at Sihwa Lake revealed females had a greater average age than the males. Many other anurans [38,39], including microhylids [40,41], also showed similar patterns of age structures. However, our study groups’ age structure, asymptotic size, and longevity were slightly lower than that of the Jeju population of *K. borealis* [23]. This might be attributed to the differences in the weather, environment, and floral and faunal communities in these localities, as found in other species [42,43]. Furthermore, the study of the Jeju population did not follow the BCM, and did not clarify the endosteal region, periosteal region, or the KL, and considered all lines as LAGs [23]. Thus, we suspect an overestimation of the age structure of the Jeju population in the previous study and recommend a proper verification.

In addition to elucidating the age structure, we also observed the relationship of age to body size and weight in the *K. borealis* population at Sihwa Lake. Our study species showed a positive correlation between age and body size in both sexes. Although there are some exceptions [44,45,46], most anurans are reported to have a positive correlation between age and body size in both sexes, like our results [39,46,47]. Additionally, we observed female-dominant sexual size dimorphism in the studied species (females were larger than the males), which is very common in anurans [48]. Although the larger sizes of females in most anurans are explained by later sexual maturity than males [37,49,50], this did not seem to suit our case. Our species reached sexual maturity at the same age: two years. Considering sexual size dimorphism as a consequence of sexual and natural selection [51], our result suggests a natural favor for larger body size in females, possibly to obtain a higher fecundity and larger offspring size [46,52]. It might also be attributed to males’ higher energy consumption (e.g., calling), and natural selection for avoiding predation risks in males (as males are more vulnerable because of their acoustic behavior).

## 5. Conclusions

In conclusion, we confirmed the formation of a single LAG in each hibernation period by rearing juveniles in the laboratory for more than a year. As hibernation is a yearly phenomenon in our species, and most of the other amphibians, we argue that LAGs are equivalent to year rings and the age of an individual. Thus, we validate skeletochronology as an appropriate method for determining the age of amphibians that undergo an annual hibernation and show a similar growth pattern to our species. To avoid overestimation, we suggest following the BCM and the proper identification of the IVL. Thus, we recommend verifying studies that did not follow the BCM to identify the IVL properly (e.g., the Jeju population of *K. borealis*). Although verification is needed for other species, we oppose an age underestimation in our species, as they did not show any endosteal resorption of LAGs. Our study offers accurate information on the life history traits and age structure of the *K. borealis* population at Sihwa Lake that may help to evaluate population dynamics in other areas, assess the causes of population decline, identify vulnerable life stages and sites, and set conservation priorities.

## Figures and Tables

**Figure 1 biology-11-00898-f001:**
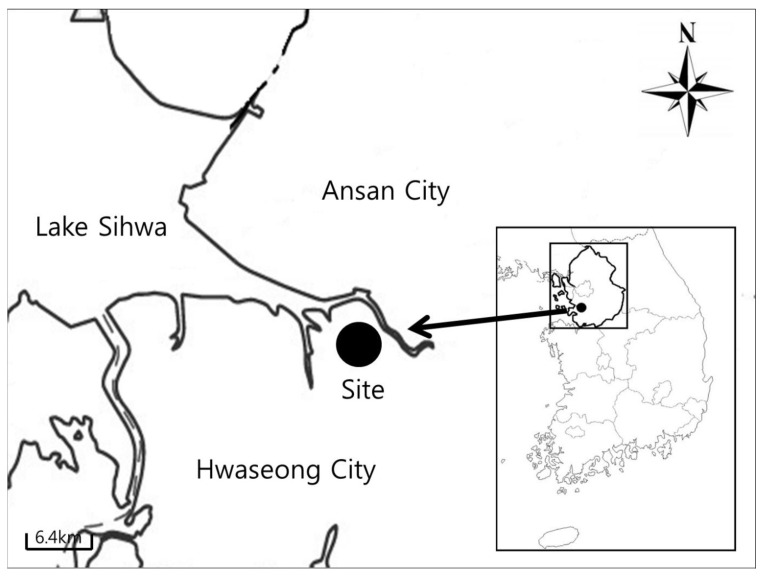
Map of study site in South Korea.

**Figure 2 biology-11-00898-f002:**
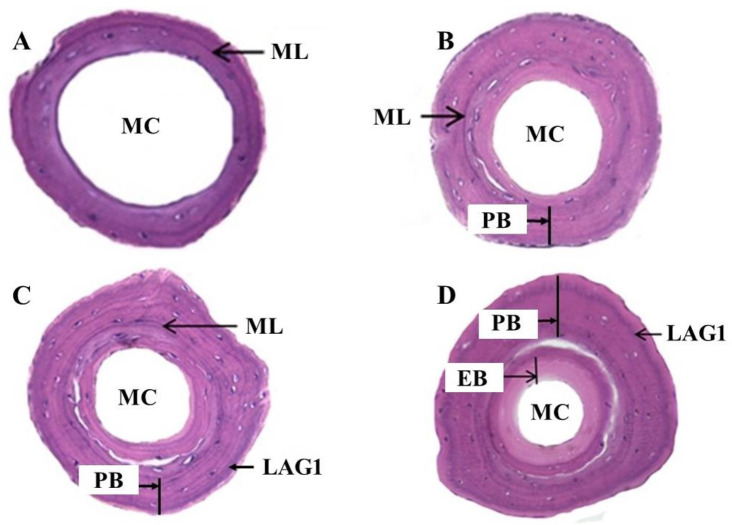
The phalangeal cross-section of laboratory-reared juvenile *K. borealis* by growth stage; (**A**) immediately after the metamorphosis; (**B**) before the first hibernation; (**C**) immediately after the first hibernation; (**D**) before the next hibernation. ML = metamorphosis line; MC = marrow cavity; PB = periosteal bone; EB = endosteal bone; LAG = line of arrested growth.

**Figure 3 biology-11-00898-f003:**
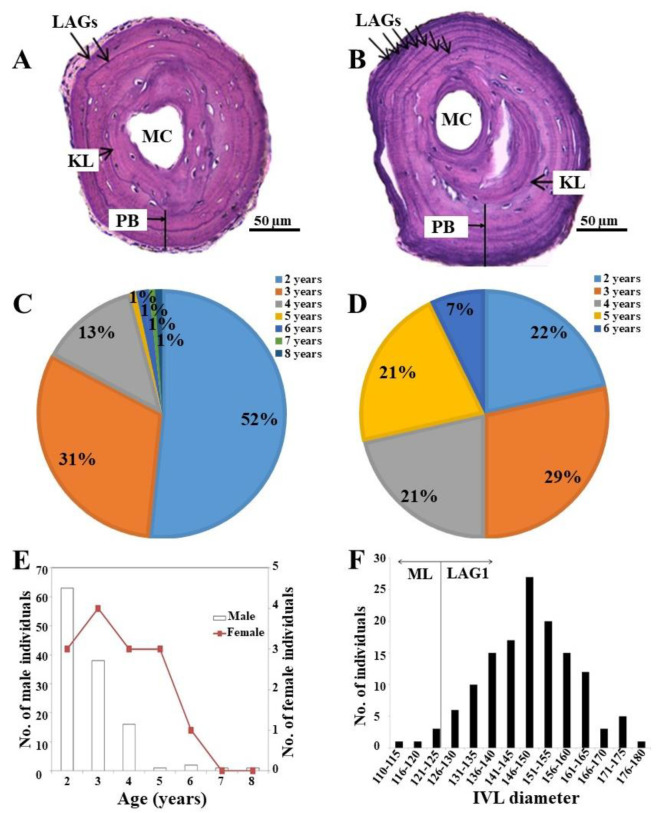
(**A**) Phalangeal cross-section of a two-year-old individual; (**B**) phalangeal cross-section of a seven-year-old individual; (**C**) percentage of age groups of the male community of adult *K. borealis* population at Lake Sihwa; (**D**) percentage of age groups of the female community of adult *K. borealis* population at Lake Sihwa; (**E**) age structure in the adult *K. borealis* population at Lake Sihwa; (**F**) IVL diameter distribution in the adult *K. borealis* population at Lake Sihwa. ML = metamorphosis line; MC = marrow cavity; PB = periosteal bone; LAG = line of arrested growth; KL = Kastschenco’s line; IVL = innermost visible line.

**Figure 4 biology-11-00898-f004:**
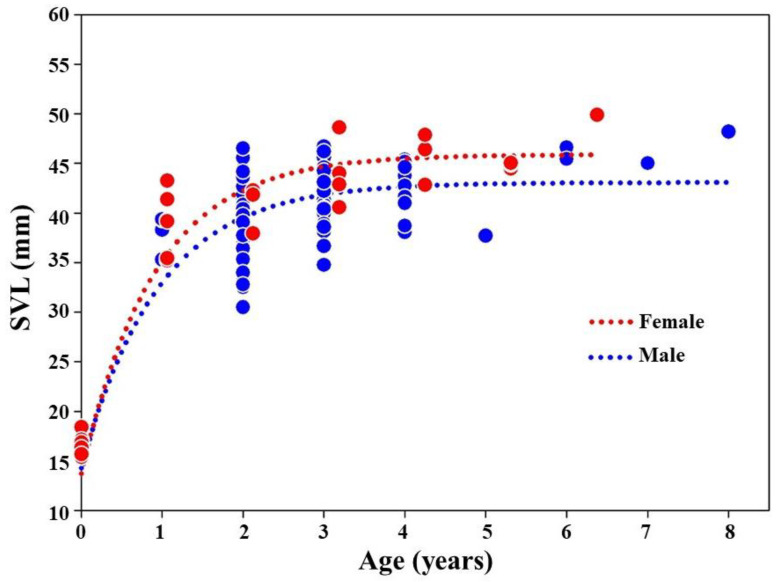
The growth curve for the adult *K. borealis* population at Lake Sihwa.

**Figure 5 biology-11-00898-f005:**
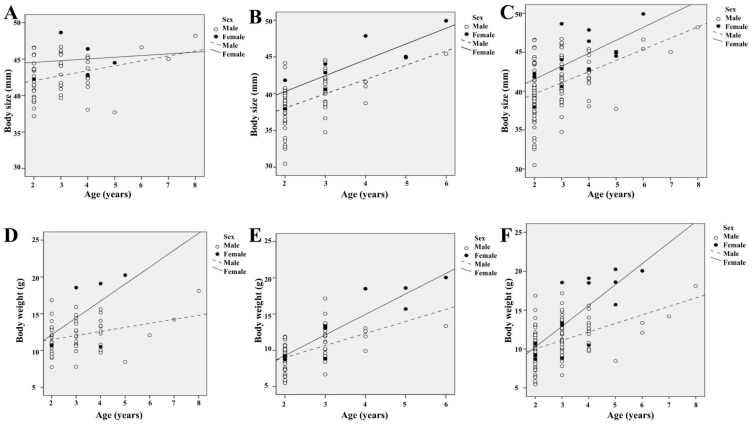
Relationship of age with body size and weight of adult *K. borealis* population at Lake Sihwa: (**A**) relationship between age and body size in the first year; (**B**) relationship between age and body size in the first year; (**C**) relationship between age and body size in the whole study period; (**D**) relationship between age and body weight in the first year; (**E**) relationship between age and body weight in the first year; (**F**) relationship between age and body weight in the whole study period.

**Table 1 biology-11-00898-t001:** The diameter values of ML and LAG1 in laboratory-reared juvenile *K. borealis*.

Life Stage	Number of Individuals	Lines	Diameter (μm)			
			Mean ± SD	Threshold *	Min	Max
Juvenile	10	ML	109.75 ± 8.64	127	96.80	123.38
		LAG1	162.90 ± 8.63	180	146.35	174.69

ML = metamorphosis line; LAG1 = first line of arrested growth. * = Mean + (2 × SD).

**Table 2 biology-11-00898-t002:** Summary of age (mean ± SD) in the adult population of boreal digging frogs (*K. borealis*) in Lake Sihwa.

Life Stage	Sex	Number of Individuals	Age (Years)		
			Mean ± SD	Range	Z
Adult	Male	122	2.75 ± 1.05	2–8	−2.845
	Female	14	3.64 ± 1.28	2–6	

Z < 0.05.

**Table 3 biology-11-00898-t003:** Maximum growth size and growth coefficient in the adult population of boreal digging frogs (*K. borealis*) in Lake Sihwa.

Sex	Number of Individuals	Maximum Growth Size (mm; Mean ± SD)	Growth Coefficient (K; Mean ± SD)
Male	122	43.08 ± 0.63	1.05 ± 0.10
Female	14	45.90 ± 1.06	1.16 ± 0.15

## Data Availability

The data presented in this study are available in Appendix A.

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
