# Peer review of "Confirmation of the Applicability of Skeletochronology and Estimating the Age Structure of Kaloula borealis (Microhylidae: Anura) at Lake Sihwa, South Korea"

_biology, 2022, doi:10.3390/biology11060898_

Round 1

Reviewer 1 Report

In the ms entitled “Confirmation of the applicability of skeletochronology and estimating the age structure of Kaloula borealis (Microhylidae: Anura) at Lake Sihwa, South Korea “, authors test a skeletochronology method with positive results in Kaloula borealis. The manuscript is scientifically well written, and the methodology is adequate. Conclusions are supported by results and the research have interest. I think the paper is publishable with minor revisions, specially those regarding the description of bioethics permissions.

-Line 32: Please provide also IUCN status

-Line 81: Please provide also IUCN status

- “Institutional Review Board Statement: The study was conducted with the permission issued by the 438 ‘Han River Basin Environmental Office’ (License Number: 2013-14).”à Usually it is also required the bioethical approval of the research institution in these cases where animals are damaged for research. The administrative permission is usually for capture and maintenance, but the bioethics committee is also required. I do not know the concrete laws of South Korea, but authors must better explain if this bioethics approval is necessary.

Reviewer 2 Report

Congratulation to the authors of the present manuscript “Confirmation of the applicability of skeletochronology and estimating the age structure of Kaloula borealis (Microhylidae: Anura) at Lake Sihwa, South Korea” for conceptualization, experimental design, study and writing. This is a well written, clear and concise manuscript with well addressed pros and cons of skeletochronology. The combination of experimental data and data from natural population gives a proper validation of the methodology. Study design is well explained and proper statistical methods were used to analyze data. Results are clear and well presented. Discussion addresses all aspects of the results and all necessary references are present. I would recommend this manuscript for publication. I only have a few minor suggestions, please find it below:

L93: put space before “-”

L96: Material and methods?

L110: replace “this species” with species name, because it is first mention in Material and methods part

L113: remove “besides” or “also”

L121: do you mean in the laboratory?

L125: add “body weight (BW)” as you mention this abbreviation later in the MS

L194-L197: rephrase this part as below to avoid redundance:

 “We used the individual age obtained by skeletochronology and SVL data in 195 generating the growth curve following von Bertalanffy's equation [28]: St = Sm – (Sm – S0)e – K(t – t0” 

L201: metamorphosis

L206-L209: rephrase this part as below to avoid redundance:

“The sexual dimorphism of the K. borealis population at Sihwa Lake was estimated following the Lovich and Gibbons (1992) Sexual Dimorphism Index (SDI) [29]: SDI = (size of larger sex/size of smaller sex) ± 1”

L217: replace “body weight” with “BW” as you use it later and introduce BW earlier (see in previous comments)

 L251-254: explain all abbreviations in the figure

 L295-300: explain all abbreviations in the figure

 L305-306: this sentence is for M&M part

 L306-315: remove values from text because they are already in the table 3, or remove table 3 if values stay in the text

L335: this sentence is for M&M part
